# Dynamically crossing diabolic points while encircling exceptional curves: A programmable symmetric-asymmetric multimode switch

Ievgen I. Arkhipov [1] ✉, Adam Miranowicz [2,3], Fabrizio Minganti[4,5], Şahin K. Özdemir [6] & Franco Nori [2,7,8] ✉

Nontrivial spectral properties of non-Hermitian systems can lead to intriguing effects with no counterparts in Hermitian systems. For instance, in a two-mode photonic system, by dynamically winding around an exceptional point (EP) a controlled asymmetric-symmetric mode switching can be realized. That is, the system can either end up in one of its eigenstates, regardless of the initial eigenmode, or it can switch between the two states on demand, by simply controlling the winding direction. However, for multimode systems with higher-order EPs or multiple low-order EPs, the situation can be more involved, and the ability to control asymmetric-symmetric mode switching can be impeded, due to the breakdown of adiabaticity. Here we demonstrate that this difficulty can be overcome by winding around exceptional curves by additionally crossing diabolic points. We consider a four-mode $\mathcal{PT}$-symmetric bosonic system as a platform for experimental realization of such a multimode switch. Our work provides alternative routes for light manipulations in non-Hermitian photonic setups.

Physical systems that are described by non-Hermitian Hamiltonians (NHHs) have attracted much research interest during the last two decades thanks to their peculiar spectral properties. Namely, such systems can possess exotic spectral singularities referred to as exceptional points (EPs). While in classical and semiclassical systems EPs are associated with the coalesce of both the eigenvalues and the corresponding eigenmodes of an NHH (thus, referred to as Hamiltonian EPs)[1,2], in quantum systems they are associated with eigenvalue degeneracies and the coalescence of the corresponding eigenmatrices of a Liouvillian superoperator (hence, Liouvillian EPs)[3]. The latter takes into account the effects of decoherence, quantum jumps, and associated quantum noise.

In addition to EPs, physical systems can also exhibit diabolic point (DP) spectral degeneracies where eigenvalues coalesce but the corresponding eigenstates remain orthogonal. Although they are often referred to as Hermitian spectral degeneracies and studied in Hermitian systems, it is well-known that DPs can emerge in non-Hermitian systems, too.

[1]Joint Laboratory of Optics of Palacký University and Institute of Physics of CAS, Faculty of Science, Palacký University, 17. listopadu 12, 771 46 Olomouc, Czech Republic. [2]Theoretical Quantum Physics Laboratory, Cluster for Pioneering Research, RIKEN, Wako-shi, Saitama 351-0198, Japan. [3]Institute of Spintronics and Quantum Information, Faculty of Physics, Adam Mickiewicz University, 61-614 Poznań, Poland. [4]Institute of Physics, Ecole Polytechnique Fédérale de Lausanne (EPFL), CH-1015 Lausanne, Switzerland. [5]Center for Quantum Science and Engineering, Ecole Polytechnique Fédérale de Lausanne (EPFL), CH-1015 Lausanne, Switzerland. [6]Department of Engineering Science and Mechanics, and Materials Research Institute (MRI), The Pennsylvania State University, University Park, PA 16802, USA. [7]Quantum Information Physics Theory Research Team, Quantum Computing Center, RIKEN, Wakoshi, Saitama 351-0198, Japan. [8]Physics Department, The University of Michigan, Ann Arbor, MI 48109-1040, USA. ✉e-mail: ievgen.arkhipov@upol.cz; fnori@riken.jp

The term DP was coined in ref. [4] referring to the degeneracies of energy levels of two-parameter real Hamiltonians. Graphically, such a DP corresponds to a double-cone connection between energy-level surfaces resembling a diabolo toy, which justifies the DP notion.

Analogously to EPs, this original definition of DPs was later generalized to the eigenvalue degeneracies of non-Hermitian Hamiltonians (see, e.g.,[5]) as DPs of classical or semiclassical systems and DPs of Liouvillians[3] in case of quantum systems. Note that quantum jumps are responsible for a fundamental difference between semiclassical and quantum EPs/DPs, and the effect of quantum jumps can be experimentally controlled by postselection[6].

EPs have been predicted and observed in different experimental platforms[1,6–22]. It seems that DPs in non-Hermitian systems have been attracting relatively less interest than EPs in recent years (see, e.g.,[1,18,23,24]). The reported demonstrations of a Berry phase (with a controlled phase shift), acquired by encircling a DP[25–27], can lead to applications in topological photonics[28], quantum metrology[29], and geometric quantum computation in the spirit of refs. [30–33]. Note that the Berry curvature (i.e., the "curvature" of a certain subspace) can be nonzero for non-Hermitian systems and, thus, can be used for simulating effects of general relativity[34–36].

The emergence of geometric Berry phases is quite common in non-Hermitian systems, but the acquired phases can be largely enhanced by encircling DPs or EPs[37–39]. Moreover, DPs and EPs are useful in testing and classifying phases and phase transitions[40,41]. For example, a Liouvillian spectral collapse in the standard Scully-Lamb laser model occurs at a quantum DP[42,43].

Recent studies on EPs have also shown that by exploiting a nontrivial topology in the vicinity of EPs in the energy spectrum can lead to a swap-state effect, where the initial state does not come back to itself after a round trip around an EP. Such phenomenon has been predicted theoretically[44,45] and observed experimentally in[21,37,46–48], while performing ‘static’, i.e., independent, measurements at various locations in the system parameter space. However, when encircling an EP dynamically, another intriguing effect can be invoked; namely, a chiral mode behavior, such that a starting state, after a full winding period, can eventually return to itself[49–52]. The latter effect stems from the breakdown of the adiabatic theorem in non-Hermitian systems[49,53]. This asymmetric mode switching phenomenon has also been experimentally confirmed in various platforms[38,54–58]. A number of studies have demonstrated the practical feasibility to observe the chiral light behavior on a pure quantum level[59] and even in a so-called hybrid mode[60], where by exploiting various measurement protocols, one can switch between the system dynamics described by a quantum Liouvillian and the corresponding classical-like effective NHH.

Other works, both theoretical[61] and experimental[62], have pointed that a crucial ingredient in detecting a dynamical flip-state asymmetry is the very curved topology near EPs. In other words, it is not necessary to wind around EPs in order to observe such phenomena. However, the dynamical contours must be in a close proximity to EPs[61].

More recently, much effort is put on studying the behavior of modes while encircling high-order or multiple EPs in a parameter space of multimode systems. Indeed, the presence of high-order or multiple low-order EPs in a system spectrum, along with the non-Hermitian breakdown of adiabaticity, can impose a substantial difficulty to manipulate the mode-switching behavior on demand[52,63,64]. That is, a system may end up only in a few states out of many regardless of the encircling direction and winding number.

In this work we demonstrate that dynamically winding around exceptional curves (ECs), whose trajectories can additionally cross diabolic curves (DCs), provides a feasible route to realize a programmable multimode switch with controlled mode chirality. We use a four-mode parity-time ($\mathcal{PT}$)-symmetric bosonic system, which is governed by an effective NHH, as an exemplary platform to demonstrate this programmable switch. At the crossing of EC and DC a new type of a spectral singularity is formed, referred to as diabolically degenerate exceptional points (DDEPs)[65]. By exploiting the presence of DDEPs in dynamical loops of the system parameter space, one can restore the swap-state symmetry, which breaks down in two-mode non-Hermitian systems. This implies that the initial state can eventually return to itself after a state flip in a double cycle. In other words, the interplay between the topologies of EPs and DPs enables one to restore (impose) mode symmetry (asymmetry) on demand. These results are valid also for purely dissipative systems (i.e., loss only systems without gain) and can be extended to arbitrary multimode systems.

## Results
### Theory
We start from the construction of a four-mode NHH, possessing both exceptional and diabolic degeneracies. For this, we follow the procedure described in[65], where one can construct a matrix, whose spectrum is a combination of the spectra of two other matrices by exploiting Kronecker sum properties. Namely, by taking two $\mathcal{PT}$-symmetric matrices

$$M_1 = \begin{pmatrix} i\Delta & k \\ k & -i\Delta \end{pmatrix}, \quad M_2 = \begin{pmatrix} 0 & g \\ g & 0 \end{pmatrix}, \qquad (1)$$

one can form a $\mathcal{PT}$-symmetric 4 × 4 non-Hermitian matrix

$$H = M_1 \otimes I + I \otimes M_2, \qquad (2)$$

where $I$ is the 2 × 2 identity matrix. Explicitly, the matrix $H$ reads

$$H = \begin{pmatrix} i\Delta & g & k & 0 \\ g & i\Delta & 0 & k \\ k & 0 & -i\Delta & g \\ 0 & k & g & -i\Delta \end{pmatrix}. \qquad (3)$$

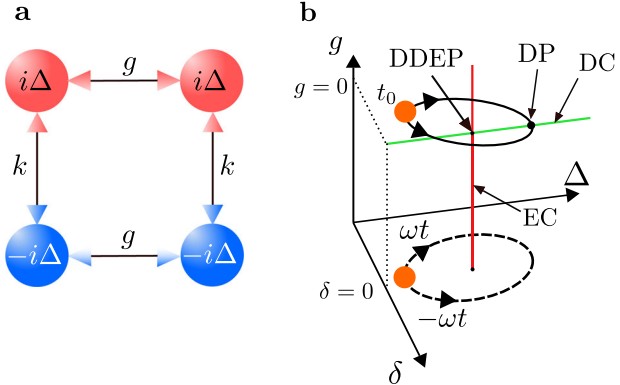

**Fig. 1 | Scheme and encircling trajectory space for a four-mode system.**
**a** Schematic representation of a four-mode $\mathcal{PT}$-symmetric non-Hermitian Hamiltonian $\hat{H}$, given in Eq. (3). The red (blue) balls represent cavities with gain (loss) rate $i\Delta$ ($-i\Delta$). Various mode couplings are depicted by double arrows. **b** The encircling trajectory is described by a loop in the 3D parameter space defined by the dissipation strength $\Delta$, perturbation $\delta$, and coupling $g$. The clockwise (counterclockwise) direction is determined by $+\omega$ ($-\omega$). The encircling starts at $t_0$ at a point in the exact $\mathcal{PT}$-phase (the orange ball). The loop winds around an exceptional curve, EC (red vertical line), determined by the condition $\Delta = 1$ and $\delta = 0$. The trajectory may cross a diabolic curve, DC (green horizontal line), at some point when $g = 0$, i.e., a diabolic point, DP. Moreover, at $g = 0$, a diabolically degenerate exceptional point, DDEP, is formed, at the intersection of EC and DC. Note that in this 3D parameter space, the DC and EC are presented as lines.

**a**

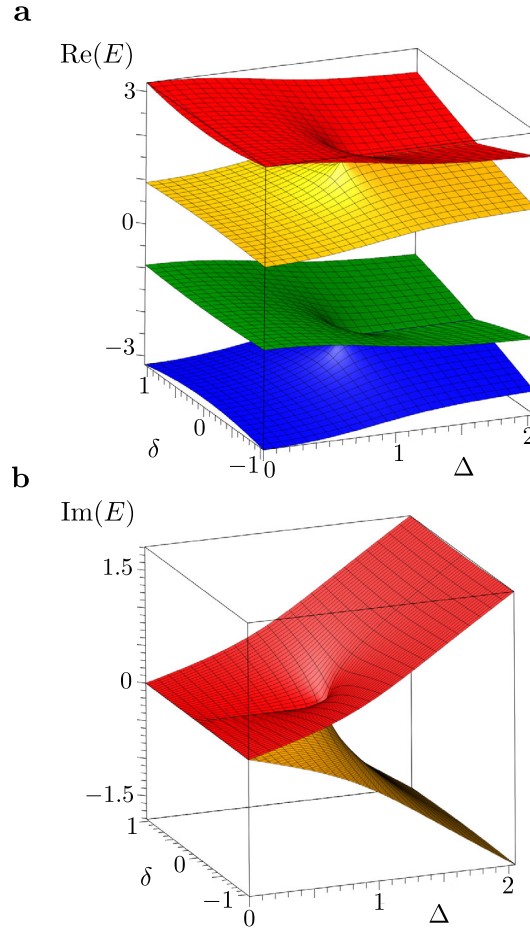

**b**

**Fig. 2 | Spectrum of a four-mode $\mathcal{PT}$-symmetric system.** Real **a** and imaginary **b** parts of the spectrum of the non-Hermitian Hamiltonian, NHH, $H(\delta)$ in Eq. (6). For real-valued energies, the spectrum of the NHH is formed by two pairs of Riemann surfaces, whereas for the imaginary-valued spectrum, those two pairs coincide. Each pair of Riemann sheets, for a given value of $g$, has a branch cut at an exceptional point determined by the conditions $\Delta = 1$ and $\delta = 0$. The system parameters are: $k = 1$ and $g = 2$.

The symbols in Eq. (3) can have various physical meanings, but in our context they may denote, e.g., coupling $(g, k)$ and dissipation $(\Delta)$ strengths in a photonic system (see the text below). The $\mathcal{PT}$-symmetry operator is expressed via the parity operator $\mathcal{P} = \text{antidiag}[1,1,1,1]$ and the time-reversal operator $\mathcal{T}$, thus, implying $\mathcal{PT}H\mathcal{PT}^{-1} = H$. The matrix $H$ can be related to a linear four-mode NHH operator $\hat{H}$, written in the mode representation, i.e.,

$$\hat{H} = \sum \hat{a}_j^\dagger H \hat{a}_k,$$

where $\hat{a}_i$ ($\hat{a}_i^\dagger$) are the annihilation (creation) operators of bosonic modes $i = 1, \dots, 4$. Such an NHH can be associated, e.g., with a system of four coupled cavities or waveguides (see Fig. 1a). A similar scheme, based on two lossy and two amplified subsystems, has been proposed in ref. [66] to generate high-order EPs but with different coupling configuration and spectrum with no DPs.

The peculiarity of such a non-Hermitian Hamiltonian $\hat{H}$ is that its eigenvalues are just sums of the eigenvalues of $M_1$ ($\pm\sqrt{k^2 - \Delta^2}$) and $M_2$ ($\pm g$)[65,67]. Namely,

$$E_{1,2,3,4} = \mp\sqrt{k^2 - \Delta^2} \mp g. \qquad (4)$$

In what follows, we always list eigenvalues in ascending order, i.e.,

$$\text{Re}(E_1) \leq \text{Re}(E_2) \leq \text{Re}(E_3) \leq \text{Re}(E_4).$$

The corresponding eigenvectors of $H$ are simply formed by the tensor products of eigenvectors of $\psi_j^{M_1}$ and $\psi_k^{M_1}$ ($j, k = 1, 2$) of the two matrices $M_1$ and $M_2$, respectively,

$$\psi_{1,2}^{M_1} = \begin{pmatrix} \pm \exp(\pm i\phi) \\ 1 \end{pmatrix}, \quad \psi_{1,2}^{M_2} = \begin{pmatrix} \pm 1 \\ 1 \end{pmatrix}, \qquad (5)$$

where $\phi = \arctan(\Delta/\sqrt{k^2 - \Delta^2})$. Namely, the eigenvector $\psi_{jk}^H = \psi_j^{M_1} \otimes \psi_k^{M_2}$ corresponds to the eigenvalue $E_{jk}^H = E_j^{M_1} + E_k^{M_2}$ of the matrix $H$[65]. The spectrum of this $\mathcal{PT}$-symmetric $\hat{H}$ has two types of degeneracies:

- a pair of second-order ECs at $k = \Delta$, determined by $\pm g$ ($g \neq 0$),
- and a pair of DCs at $g = 0$, defined by $\pm\sqrt{k^2 - \Delta^2}$.

**System dynamics in modulated parameter space**

In order to implement the dynamical winding around ECs, one may apply a perturbation $\delta(t)$ to the NHH $\hat{H}$ in the following form:

$$\hat{H}(\delta) = \begin{pmatrix} i\Delta(t) + \delta(t) & g(t) & 1 & 0 \\ g(t) & i\Delta(t) & 0 & 1 \\ 1 & 0 & -i\Delta(t) & g(t) \\ 0 & 1 & g(t) & -i\Delta(t) - \delta(t) \end{pmatrix}, \qquad (6)$$

where we set $k = 1$, i.e., the coupling $k$ determines a unit of the system energy. The time-dependent parameters are:

$$
\begin{aligned}
\Delta(t) &= 1 + \cos(\omega t + \phi_0), \\
g(t) &= g_0 \sin^2(\omega t/2 + \phi_0/2), \\
\delta(t) &= \sin(\omega t + \phi_0).
\end{aligned}
\qquad (7)
$$

The angular (winding) frequency is $\omega = 2\pi/T$, with period $T$, and an initial phase $\phi_0$. The perturbation $\delta$ can play the role of the frequency detuning in the first and fourth cavities. Other choices of perturbation are also allowed, although they can lead to a different energy distribution in the perturbed parameter space.

The energy spectrum of $H(\delta)$ consists of two pairs of Riemann sheets. For real-valued energies, these pairs may or may not intersect, depending on the system parameters, as shown in Figs. 2a and 3. For imaginary-valued energies, on the other hand, these pairs always coincide, as follows from Eq. (4) (see also Fig. 2b). Though the chosen perturbation lifts the $\mathcal{PT}$-symmetry of the NHH in Eq. (3), the NHH $\hat{H}(\delta)$ still possesses the chiral symmetry $\mathcal{C}\hat{H}(\delta)\mathcal{C}^{-1} = -\hat{H}(\delta)$, where $C$ is the Hermitian operator satisfying $\mathcal{C}^2 = 1$, expressed via the antidiagonal matrix $\mathcal{C} = \text{antidiag}[1, -1, -1, 1]$. For this chiral symmetry one always has: $E_k = -E_{5-k}$, with $k = 1, \dots, 4$ (see Fig. 2).

In order to determine the time evolution of a wave function $\psi$, during a dynamical cycle, we solve the time-dependent Schrödinger equation

$$i\frac{\partial\psi(t)}{\partial t} = \hat{H}(t)\psi(t). \qquad (8)$$

Here, we focus solely on the mode switching behavior in the stable exact $\mathcal{PT}$-phase, where the eigenvalues $E_k$ are real-valued, thus, representing propagating fields without losses. That is, the dynamical encircling starts in the exact $\mathcal{PT}$-phase (i.e., $\Delta < 1$).

The basic idea of the proposed scheme for the controlled chiral mode switching can be described as follows. The encircling loop moves in a 3D-parameter space spanned by the dissipation rate $\Delta$, the detuning perturbation $\delta$, and the coupling $g$ (see Fig. 1b). Encircling one of the ECs (e.g.,$+g$) automatically ensures that EC ($-g$) is also encircled

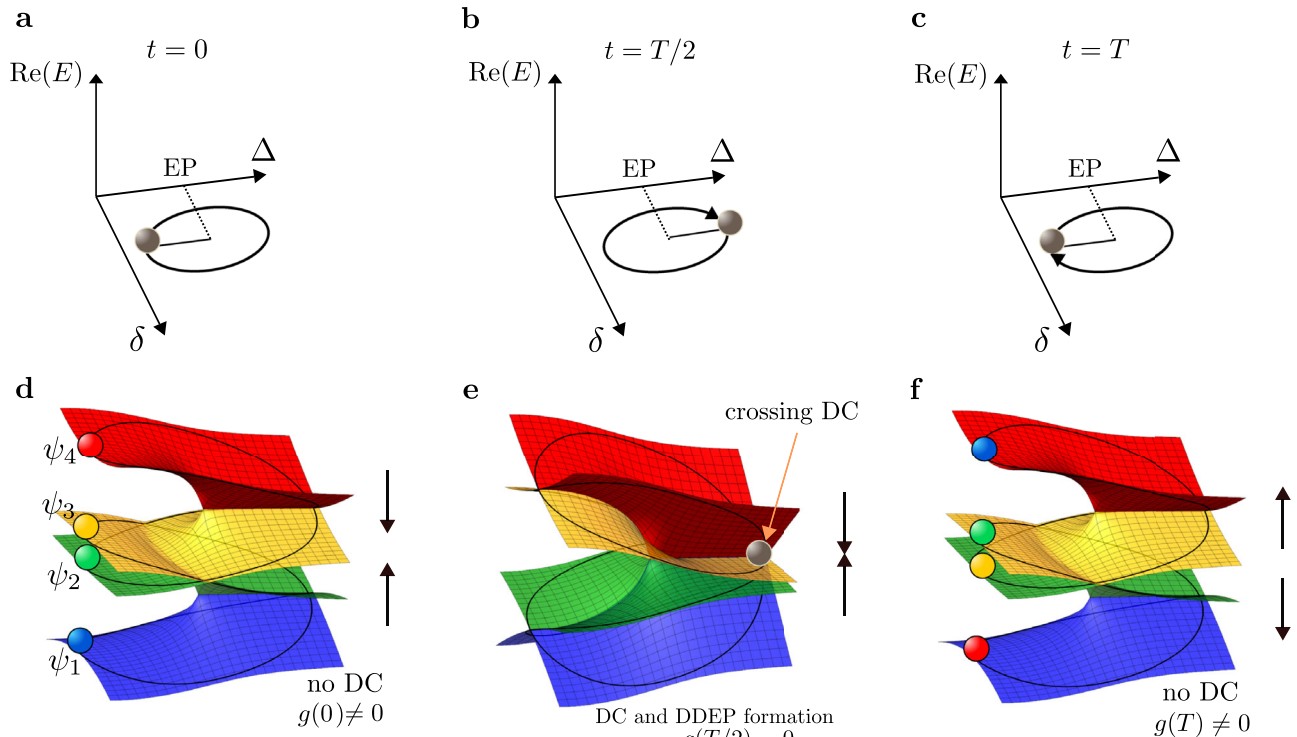

**Fig. 3 | Encircling an exceptional curve, EC, by crossing a diabolic curve, DC, in a four-mode system described by an NHH in Eq. (6). a–c** The encircling trajectory projected on the 2D parameter space is defined by the dissipation strength $\Delta$ and the perturbation $\delta$ at different stages of the encircling process. The vertical axis is the real-valued energy Re($E$). The grey ball represents an evolving system eigenmode. **d–f** Real-valued energy Riemann sheets at corresponding stages of the encircling process, whose axis are the same as in panels **a–c**. **a, d** Initial state at $t = 0$: two separated exceptional points, EPs. At one of the EPs there is branch cut between the red and yellow Riemann sheets, and at the other EP there is a branch cut between the green and blue sheets. Four different eigenmodes $\psi_{1,2,3,4}$ are depicted as colored balls. **c, e** When the encircling trajectory crosses the DC in the broken PT-phase at the half period, a diabolically degenerate exceptional point (DDEP) is formed that connects various energy Riemann sheets. The presence of a DC is indicated by the intersection of two pairs of planes (the red and yellow sheets, and the green and blue sheets). The encircling trajectories (black solid curves) cross the DC at some point, i.e., a DP. Trajectories of all eigenmodes coincide when crossing a DC and are represented by the grey ball. **c, f** Final state at $t = T$, with the same system parameters as for the initial state: the eigenmodes are permuted compared to the initial modes, and that shuffling depends on the direction of the encircling and whether the encircling trajectory passes through the DC or not. The DC crossing is induced by the appropriate time modulation of the coupling $g$ (see the main text for more details).

due to the system symmetry. For a given fixed value $g$, there is a distance $|2g|$ between two EPs, belonging to the two ECs (see, e.g., Fig. 3b).

The initial point ($t_0 = 0$), from which the encircling trajectory starts, is located in the exact $\mathcal{PT}$-phase, i.e., $\phi_0 = \pi$ (see Figs. 1b and 3a, d). The winding process can be performed in the clockwise ($+\omega$) or counterclockwise ($-\omega$) direction. By appropriately modulating $g(t)$, one can make the encircling trajectory to pass through the DC at some point, (i.e., a DP), when $g = 0$ in the broken $\mathcal{PT}$-phase ($\Delta > 1$) (as shown in Fig. 1b). A single dynamical loop, thus, corresponds to the splitting-crossing-splitting behaviour for the two pairs of the Riemann energy sheets (as shown in panels **d–f** in Fig. 3). The intersection of the sheets occurs at the DC.

Figure 3d depicts the initial state at $t = 0$ when $g \neq 0$ and the spectrum of the NHH $\hat{H}$ consists of two disconnected pairs of Riemann sheets (for real-valued $E$), where each pair is formed around a second-order EP (with characteristic branch cuts) (see Fig. 3d). As mentioned above, depending on the coupling $g \neq 0$, these Riemann pair sheets can cross (as shown in Fig. 3e). The state can be initialized in one of the four different eigenmodes in the exact $\mathcal{PT}$-phase.

**Winding an exceptional curve without crossing diabolic points**

If $g \neq 0$ is either modulated such that the two separated pairs of the real-valued Riemann sheets do not cross at the DC (as presented in Fig. 3d) or it is kept fixed, then the dynamical loop is similar to the case of two independent two-mode systems, for which the dynamical winding around an EP results in the well-known two-mode asymmetric

switching[54]. This means that, in this specific case, only the eigenmodes $\psi_1 \leftrightarrow \psi_3$ and $\psi_2 \leftrightarrow \psi_4$ which belong to the separated pairs of Riemann sheets[59] are swapped. For later convenience, we recall the known results for the mode switching combinations in such disconnected two two-mode systems ;

$$
\begin{aligned}
\circlearrowright : \quad &\psi_1 \longrightarrow \psi_3, \quad \psi_3 \longrightarrow \psi_3, \\
\circlearrowright : \quad &\psi_2 \longrightarrow \psi_4, \quad \psi_4 \longrightarrow \psi_4,
\end{aligned}
\tag{9}
$$

for clockwise winding, and

$$
\begin{aligned}
\circlearrowleft : \quad &\psi_1 \longrightarrow \psi_1, \quad \psi_3 \longrightarrow \psi_1, \\
\circlearrowleft : \quad &\psi_2 \longrightarrow \psi_2, \quad \psi_4 \longrightarrow \psi_2,
\end{aligned}
\tag{10}
$$

for counterclockwise winding, respectively. Equations (9) and (10) show the possibility to perform symmetric and asymmetric mode switching for the two-mode systems. One can always swap between $\psi_1$ and $\psi_3$, as well as $\psi_2$ and $\psi_4$ by simply changing the encircling direction corresponding to symmetric switching. For example, a system starting at $\psi_1$ will end at $\psi_3$ when encircling in the clockwise direction and the system will return back to $\psi_1$ when encircling direction is reversed. One can achieve asymmetric mode transfer if encircling is performed in a fixed direction. For example, a system at $\psi_1$ will end up at $\psi_3$ when encircling in the clockwise direction and the system will stay at $\psi_3$ if encircling is further continued in the clockwise direction. This implies that once the states are swapped they do not switch anymore, if

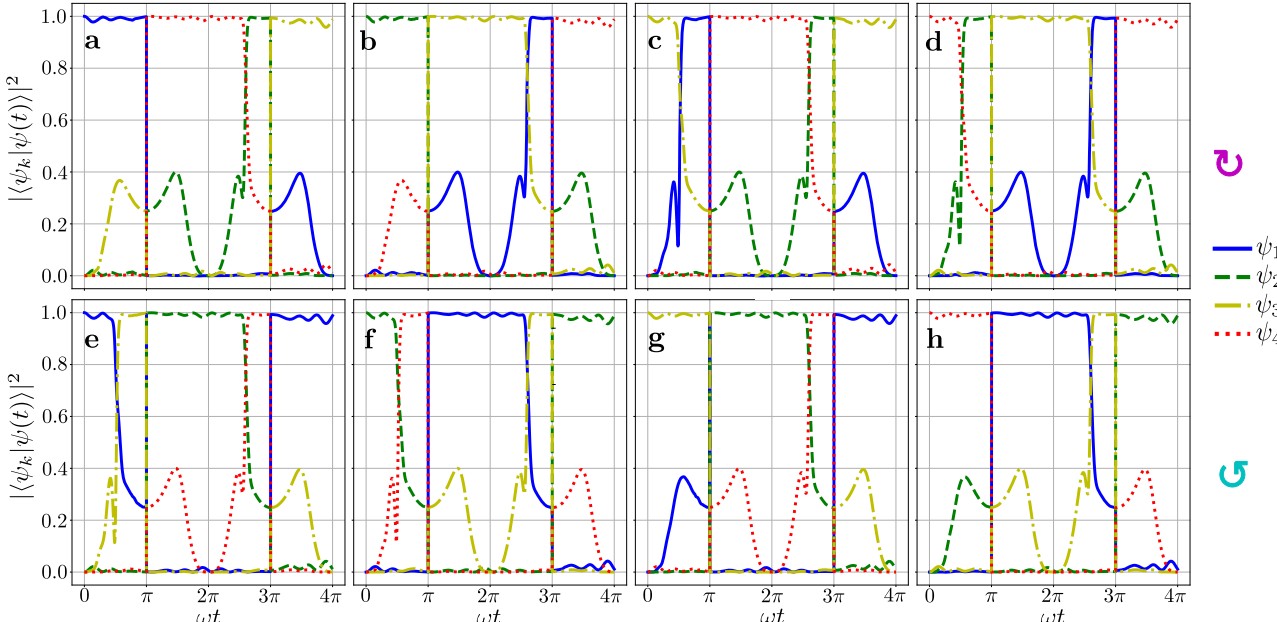

**Fig. 4 | Fidelity $|\langle\psi_k|\psi(t)\rangle|^2$ of the NHH eigenstate $\psi_k$ at time $t$ and the time-evolving state $\psi(t)$ during a double period $2T$.** The initial eigenmodes $\psi_k, k = 1,...,4$, are located in the exact $\mathcal{PT}$-phase (see also Fig. 3b). Clockwise (panels **a–d**) and counterclockwise (panels **e–h**) encircling directions. Depending on the winding direction and the number of times the loop encircles the exceptional curve, EC, with the diabolic curve, DC, crossing, one can realize various mode-switching combinations. Mode-switching combinations, illustrated here, are summarized in Table 1. These panels also reveal the occurrence of NATs, which, for given system parameters, take place either at angles $\omega t \approx \pi/2$ or $\omega t \approx 5\pi/2$. The DC crossing corresponds to phases $\omega t = \pi, 3\pi$ (see the main text for more details). The system parameters are: $\phi_0 = \pi$, $\omega t = \pi t/40$, and $g_0 = 0.5$. For better readability of the system dynamics shown here, at each moment of time the states are normalized, giving thus the fidelity range between zero and one. Otherwise, due to the non-Hermiticity, the norm of the evolving state varies.

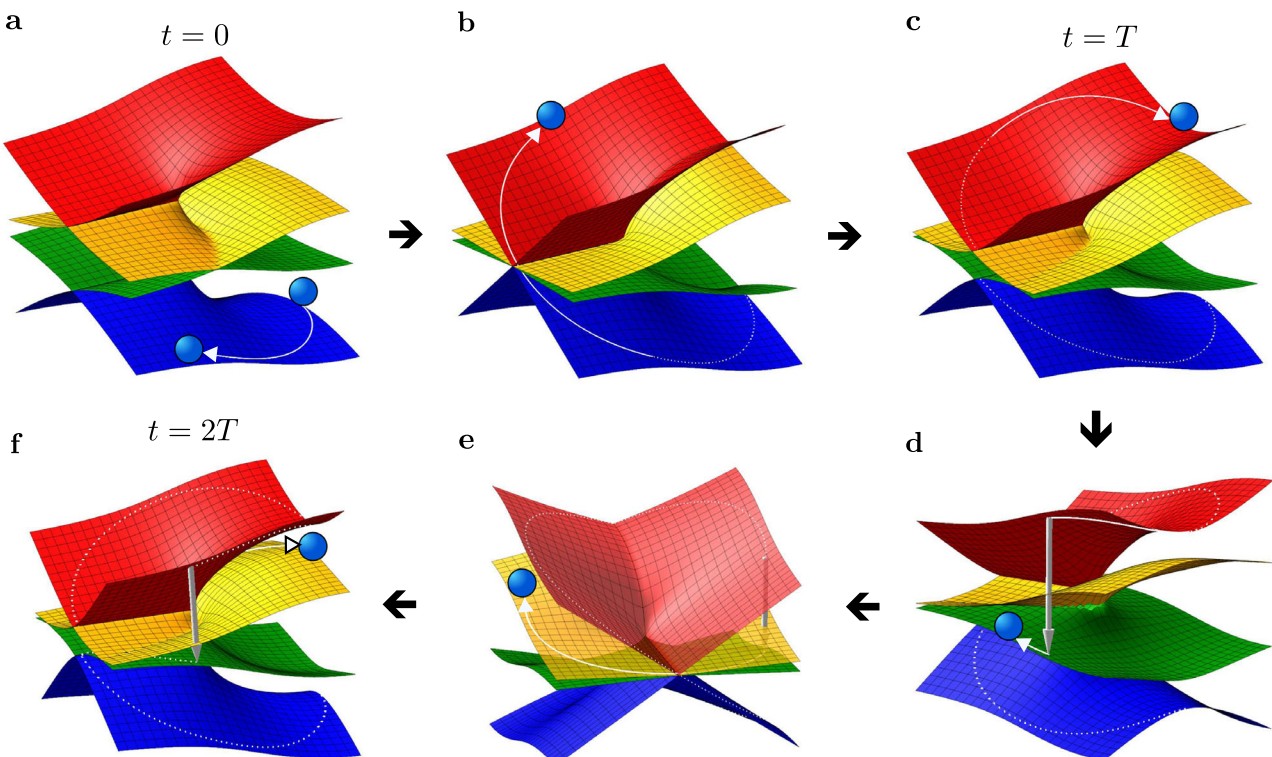

**Fig. 5 | The time evolution of the system state at different times, according to Fig. 4**a. **a** Initial state $t = 0$, **b** $T/2 < t < T$, **c** $t = T$, **d** $T < t < 3T/2$, **e** $3T/2 < t < 2T$, **f** final state $t = 2T$. See also the main text for details.

**Table 1 | Programmable four-mode switch, a summary of Fig. 4**

| Final<br>Initial | $\psi_1$ | $\psi_2$ | $\psi_3$ | $\psi_4$ |
|---|---|---|---|---|
| $\psi_1$ | 2↻ | 1↺ | 2↺ | 1↺ |
| $\psi_2$ | 1↺ | 2↺ | 1↺ | 2↺ |
| $\psi_3$ | 2↻ | 1↺ | 2↺ | 1↺ |
| $\psi_4$ | 1↺ | 2↺ | 1↺ | 2↺ |

By initializing a state in one of the system eigenmodes $\psi_k$ (first column), one can switch to any final eigenstate $\psi_j$ (first row) of the system by appropriately choosing an encircling trajectory, which winds around the exceptional curve, EC, and always traverses the diabolic curve. The order and meaning of the values and symbols in each cell is the following: the values 1, 2 denote the number of times one winds around the EC and the symbols ↺ and ↻ denote counterclockwise and clockwise encircling directions, respectively (see Fig. 4).

encircling is continued in the same direction, breaking thus flip-state symmetry. This asymmetry stems from the non-adiabatic transitions (NATs), which occur when adiabatic evolution breaks down and the system discontinuously jumps between two states[54].

## Winding an exceptional curve by crossing diabolic points

Interestingly, in order to realize any desired mode switching combination, and thus also to restore state-flip symmetry, one just needs to ensure that an encircling trajectory crosses the DC for $g = 0$ at times $t = T/2$ or $T$, depending on whether the loop goes once or twice around the EC, respectively (as shown in Fig. 3e). If the dynamical cycle crosses the DC, the two EPs coincide, forming thus a DDEP[65]. When this happens, the eigenstates can move across the four different Riemann sheets (see Fig. 3e), and any desired final state can be obtained. Moreover, after such an induced symmetric state swap, one can additionally impose the asymmetric state switching by continuing encircling EPs but without crossing the DC.

In order to better understand the system dynamics and the interplay between EC, DC, and NATs, we plot a graph for the fidelity $|\langle \psi_k | \psi(t) \rangle|^2$ of the NHH eigenstates at times $t$ ($\psi_k$) and the time-evolving ($\psi(t)$) states in Fig. 4. The symbol $\langle \cdot | \cdot \rangle$ here denotes the Hilbert inner product of two states.

Let us focus on the description of the panel a of Fig. 4 and its accompanying spectral plot in Fig. 5. At time $t = 0$, the system is initialized in the state $\psi = \psi_1$ (blue curve in Fig. 4a, shown also as a blue ball in Fig. 5a). By encircling in the clockwise direction, the state $\psi$ remains on the corresponding Riemann sheet $E_1$ for times $t < T/2$. At $t = T/2$, the state crosses DC, and is transferred to the Riemann sheet $E_4$ (see also Fig. 5b). Note that the Riemann sheets $E_1$ and $E_4$ are completely disconnected when $g \neq 0$. Thus, after the full cycle $t = T$, the initial state is switched to $\psi_1 \longrightarrow \psi_4$ (see also Fig. 5c). By continuing the winding process, the system experiences the NAT approximately at time $t \approx 5\pi/2$ (Fig. 4a and Fig. 5d). This NAT corresponds to a discontinuous jump of the state $\psi(t)$ from the $E_4$ to the $E_2$ Riemann surface. At $t = 3\pi$, the DC is crossed once more, and the $\psi(t)$ switches to the state $\psi_3(t)$ (Fig. 5e). Thus, after completing the full dynamical double loop $t = 2T$, the final state becomes $\psi(2T) = \psi_3$ (Fig. 5f).

All the panels in Fig. 4 can be described and visually represented in the same way we discussed above for panel 4a. Note that there are also trajectories which assume two NATs occurring during the time evolution (see panels Fig. 4c–f). The dynamical loops with two NATs correspond to the cases when after the double period the system returns to its initial state. The observed NATs here correspond to the NATs occurring in the $\mathcal{PT}$-symmetric dimers (when two pairs of Riemann sheets are decoupled), stemming from the interplay between loss and gain[54].

We summarize the results shown in Fig. 4 also in Table 1. This table combined together with Eqs. (9) and (10) serves as a protocol for the realization of a programmable symmetric-asymmetric mode switching

in the $\mathcal{PT}$-symmetric four-mode bosonic system. According to Table 1, by choosing an appropriate winding number and direction, one can always swap between different modes on demand, realizing thus a symmetric mode switch when traversing the DC. On the other hand, exploiting the asymmetry, expressed via Eqs. (9), (10), one can force the system to end up in one of its eigenstates regardless of the initial mode. For instance, to ensure that after two dynamical cycles in the same direction the final state always be $\psi_4$ one can perform the following protocol. First, encircling in the clockwise direction without crossing the DC will bring the system's state either to $\psi_3$ or $\psi_4$ after the period $T$, regardless of the initial state, in accordance with Eq. (9). If one detects $\psi_4$ at $t = T$, the protocol is completed because after the system will always stay in this state provided that any subsequent clockwise encircling does not cross the DC. Otherwise, that is if state $\psi_3$ is detected at $t = T$, clockwise encirclement continues with the DC crossing which results in the desired mode $\psi_4$ at $t = 2T$, according to Table 1. A similar procedure can be implemented for any system eigenmode when winding in a given direction.

Interestingly, the presented switching mechanism also allows to restore the flip-state symmetry, which is otherwise broken without the mode coupling modulation [see Eqs. (9), (10)]. Indeed, according to Fig. 4, by periodically traversing the DCs the mode nonreciprocity is eliminated for a given winding direction. For example, the system periodically switches between states $\psi_1 \leftrightarrow \psi_2$ ($\psi_3 \leftrightarrow \psi_4$) in the counterclockwise (clockwise) direction. These results contrast with systems with high-order EPs, where arbitrary mode switching is hard to realize and the chiral mode behavior cannot be controlled[56].

## Discussion

Our analysis shows that the mode switching presented is resilient to various forms of perturbations. For instance, as Fig. 4 indicates, changing the starting point of winding does not affect the results. The same applies when perturbing gain, loss, or mode coupling. Moreover, the system is robust to small perturbations in the mode coupling $g \rightarrow g + \epsilon$, for $\epsilon \ll g$, at times $t = T/2, T$. The latter fact can be understood as the diabatic evolution (on the scale of $\epsilon$) of the state in the vicinity of the DC, which enables the state to transfer to another energy surface even though the state does not cross the DC.

Note that the winding speed cannot be arbitrary. Winding too fast is similar to diabatic evolution and will bring the system to a final state which is superposition of the eigenstates. On the other hand, if winding is too slow, various NATs can start playing more vivid role for longer times which may affect the final state.

In this study we have focused on the photonic four-mode $\mathcal{PT}$-symmetric system, assuming that the gain and loss are balanced. The natural question arises whether the results are also applicable to purely passive $\mathcal{PT}$-symmetric setups. Our numerical analysis implies that one can indeed extend the obtained findings to passive systems, though it may impose certain constrains on the system parameters when decreasing the gain/loss ratio (see Supplementary Note 1). This can be useful for quantum information processing or low-power classical applications. For instance, effective passive NHHs can be realized in quantum systems exploiting various procedures such as post-selection[11,68,69] or dilation[59]. Concerning classical optical platforms, one can use a photonic system of coupled toroidal microcavities[9], or, for instance, a set of coupled waveguides, similar to that used in[54]. We also note that our findings can also be extended to anti-$\mathcal{PT}$-symmetric systems too. In this case, the role of freely propagating fields in $\mathcal{PT}$-symmetric setups can be played by dissipating fields in the corresponding anti-$\mathcal{PT}$-symmetric systems[57].

Our results are not limited to the four-mode system considered here but can be extended to arbitrary multimode photonic systems. By utilizing the method described in[70] one can construct various $N$-mode photonic setups with similar spectral structure characterized by separated pairs of Riemann surfaces as in Fig. 2. In the Supplementary

Note 2, we show the implementation of a programmable symmetric-asymmetric mode switching for in an eight-mode $\mathcal{PT}$-symmetric system.

In conclusion, by exploiting both diabolic and exceptional degeneracies in a non-Hermitian system, one can realize a programmable symmetric-asymmetric multimode bosonic switch by dynamically traversing a DC while encircling an EC. We have illustrated our results using a four-mode $\mathcal{PT}$-symmetric system and have also discussed their extension to arbitrary multimode systems. Our findings are not limited to free propagating fields in $\mathcal{PT}$-symmetric systems, but are also applicable to linear passive $\mathcal{PT}$-symmetric and anti-$\mathcal{PT}$-symmetric setups. Our work opens new perspectives for light manipulations in photonic systems.

## Data availability
The data that support the findings of this study are available from the corresponding author upon reasonable request.

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

## Acknowledgements

A.M. is supported by the Polish National Science Centre (NCN) under the Maestro Grant no. DEC-2019/34/A/ST2/00081. S.K.O. acknowledges support from Air Force Office of Scientific Research (AFOSR) Multidisciplinary University Research Initiative (MURI) Award on Programmable systems with non-Hermitian quantum dynamics (Award no. FA9550-21-1-0202) and the AFOSR Award FA9550-18-1-0235. F.N. is supported in part by: Nippon Telegraph and Telephone Corporation (NTT) Research, the Japan Science and Technology Agency (JST) [via the Quantum Leap Flagship Program (Q-LEAP), and the Moonshot R&D Grant Number JPMJMS2061], the Asian Office of Aerospace Research and Development (AOARD) (via Grant no. FA2386-20-1-4069), and the Foundational Questions Institute Fund (FQXi) via Grant no. FQXi-IAF19-06.

## Author contributions

I.A. conceived the project and performed calculations. I.A., A.M. and F.M. interpreted the results. Ş.K.Ö. and F.N. supervised the project. All authors contributed in writing the manuscript.

## Competing interests

The authors declare no competing interests.
