## [Peer review file · Nature Communications]

REVIEWER COMMENTS

Reviewer #1 (Remarks to the Author):

In the manuscript titled “Dynamically encircling an exceptional curve by crossing diabolic points: A programmable multimode switch”, the authors proposed the design of a four-mode non-Hermitian switch using the topological energy surface structures around an exceptional curve. The key to realizing the switch functionality is that they introduce a diabolic point into the non-Hermitian system. In each encircling loop, the diabolic point will be crossed once. In this way, an input mode can be converted into any desired mode by choosing a proper winding number (1 or 2 in this work) and a winding direction (i.e., clockwise or counter-clockwise). I will first talk about why in my opinion this work deserves publication although it is a theoretical work without experimental verification. The pioneering work on the dynamical encirclement of an exceptional point was published in [Nature 537, 76 (2016)], where the authors studied a two-state non-Hermitian system and a chiral mode switching phenomenon was found due to the non-adiabatic transitions between eigenmodes in non-Hermitian systems. Such breakdown of adiabaticity cannot be avoided in the dynamical process so that even multi-state systems have been studied in various works, we still cannot realize mode switching on demand as that in the stroboscopic case. After studying this work, I realize that it may be because that previously we focus on an encircling process in which two system parameters are changing and therefore we have an unchanged energy surface. In this work, the authors introduced a third varying parameter (i.e., the term “ g ”) so that the energy surface is also changing when the state evolves on it. In this way, by introducing a diabolic point which can connect two “sub-systems”, the varying energy surface allows a desired mode switching functionality. I believe that such mode switching behavior cannot be realized using previous non-Hermitian design principles. I think this work adds something very important and interesting to this research field and the design idea (i.e., a varying energy surface) will inspire researchers to find new non-Hermitian physics and applications. Therefore, I would like to support the publication of this work and I give some comments below for the authors’ consideration.

(1) The system proposed in this work is quite complex compared to previous systems with an unchanging energy surface. Therefore, it would be better to add some discussions on the non-adiabatic transitions. I assume there are at most two non-adiabatic transitions, e.g., in a process where both the initial and final mode are ψ_1 . Since different switching processes can have different number of non-adiabatic transitions, such information may help the readers better understand the underlying physics.

(2) The system is a PT-symmetric system with gain and loss. What about the phenomenon in a passive system with loss only, as such passive system may be realized in potential experiments. As there may be multiple non-adiabatic transitions in the proposed system, i.e., the state has more chances to evolve on the higher-loss energy sheet, the transmission may be very low for potential applications.

(3) What symmetry governs that the system always has two energy levels with the same lowest loss (or highest gain)?

(4) In a conventional system where “ g ” is a constant, the proposed programmable switching functionality cannot be realized, since given an initial state and winding direction, one winding loop and

two winding loops will result in the same final mode. The authors may add some discussions or a table on this point, which can emphasize the role of the diabolic point in realizing the phenomenon and also help readers intuitively understand the breakthrough of this work.

(5) In the current system, the term “ g ” is required to be zero at $\omega t = \pi$, in order to reach the diabolic point. What happened if “ g ” is very small but not zero at that point? Can we still have the same mode switching phenomenon?

(6) The labels “ a ” to “ f ” in Fig. 3 seem to be not in the same order as that mentioned in the main text. The authors should check this point.

Reviewer #2 (Remarks to the Author):

The manuscript presents theoretical studies of a programmable multimode switch based on dynamically encircling an exceptional curve while crossing diabolic points in non-Hermitian systems. A distinguishing aspect here is the proposed-a new programmable mode switch at a four-states PT-symmetric system with a diabolically degenerate exceptional point by breaking adiabaticity. The topic is interesting to broad communities of theoretical and applied physics and photonics researchers dealing with PT-symmetric systems.

There are a number of questions and comments:

Given that the fabrication imperfections are omnipresent in the experiment, how is such a system robust for coupling, gain, and loss perturbation? Such information definitely will elevate the work.

The authors claim that the results for the four-states system can be extended to any even-multimode PT system with a set of m second-order ECs accompanied by the corresponding pairs of DCs. In this respect, and given that the work is purely theoretical, it would be beneficial to present an example of the higher-states system in supplementary materials.

The photonics platform seems to be the perfect candidate to realize such a switch. Can authors propose one of the specific photonic systems and physical parameter ranges for such realization?

In general, the results are of high technical quality, and the work can be suitable for publication after a revision.

Reviewer #3 (Remarks to the Author):

In this work, the authors describe a tuneable switch between 4 modes with gain and loss, which relies on the existence of certain features (diaboloic points and exceptional curves (EC)) in the complex spectrum of those modes. In particular, it is shown that under certain conditions, each of the initial eigenmodes can be mapped onto any other mode by encircling the EC once or twice and clockwise or counter-clockwise.

While this analysis extends previous works on encircling exceptional points (EP) and will be of interest for the targeted audience, I have several reservations about a publication of this manuscript in a broad and high-impact journal such as Nature Communications:

1) Over the past ~5 years there has been a considerable interest in the dynamical encircling of EP and in the resulting asymmetric mode switching arising from a break-down of adiabaticity in non-Hermitian systems. These effects are somewhat counterintuitive when viewed from the perspective of Hermitian quantum mechanics, but can be easily understood in terms of coupled modes with varying gain and loss coefficients. Also, in contrast to what is often claimed in the literature, the existence of exceptional points or other 'topological' features in the eigenvalues spectrum is not essential to observe these effects. While encircling an EP leads to the correct alternation between loss, gain and coherent interactions, the same evolution can be obtained by simply switching on and off these contributions in the correct order.

In the current example, the existence of a diaboloic point (which one could also simply call a level crossing) switches off the coupling between some modes, which would otherwise be mixed and lead to a more complicated evolution. This makes it convenient to achieve a desired coherent mixing or non-mixing of modes, but it doesn't introduce a new fundamental principle.

2) Identifying a way to achieve a tuneable four-mode switch through the encircling of EC is an interesting academic exercise, but doesn't seem to have any practical value. There are much simpler ways to realize the same capabilities, for example, by turning on and off coherent couplings. The current dissipative implementation has the additional drawback, that it cannot be used for quantum information processing or low-power classical applications.

3) The discussion around Fig. 3 is extremely difficult to follow and it is very hard to get any intuition out of it. It might be a lot easier to understand the evolution by plotting, for example, the real parts of the eigenvalues along the parameter trajectory, $\text{Re } E_k(t)$, and indicating the instantaneous loss/gain rates of each eigenstate by its color or similar means. In any case one should think about how to make this discussion more clear.

4) There is no discussion about the imaginary part of the eigenvalues, which is responsible for the non-adiabatic transitions between the eigenvalues sheets. In the encircling of a single EP, this is the most essential part in the dynamics, since the system will always end up in the mode with maximal gain (or minimal loss). What is the role of non-adiabatic transitions in the current setting or how can it be that these non-adiabatic effects don't appear?

5) Related to 4), in the plots in Fig. 4, all the overlaps between the time-evolved wavefunction and the initial eigenmodes are of order 1. However, during the evolution, the different modes experience gain and loss and one expects that the amplitudes are amplified or damped by many orders of magnitudes. Why does this not show up in these plots?. How sensitive is the switching behavior with respect to variations in ϕ_0 and T ?

In summary, I cannot recommend a publication of this manuscript in Nature Communication, both in terms of its overall impact, but also because crucial aspects concerning the dissipative mode evolution are missing in the current discussion.

Re: NCOMMS-22-44463

ORIGINAL TITLE: Dynamically encircling an exceptional curve by crossing diabolic points:
A programmable multimode switch

REVISED TITLE: Dynamically crossing diabolic points while encircling exceptional curves:
A programmable symmetric-asymmetric multimode switch

AUTHORS: I. I. Arkhipov, A. Miranowicz, F. Minganti, Ş. K. Özdemir, and F. Nori

Dear Dr. Matricardi:

Thank you very much for your message and consideration of our manuscript for publication in Nature Communications.

We also thank the Referees for their careful reading of our work and for raising a number of insightful comments and suggestions.

In the revised version of the manuscript, and in this reply, we address point-by-point all the Referees comments and critique. In particular, to adequately address some comments of the Referees, we have also modified the article title, as indicated above.

We believe that this revised version, with the improved presentation of our results, is now suitable for publication in Nature Communications.

We thank you again for your time and consideration.

Kind regards,

the authors

***** REPORT OF THE REFEREE 1 *******REFEREE'S COMMENT 1**

In the manuscript titled “Dynamically encircling an exceptional curve by crossing diabolic points: A programmable multimode switch”, the authors proposed the design of a four-mode non-Hermitian switch using the topological energy surface structures around an exceptional curve. The key to realizing the switch functionality is that they introduce a diabolic point into the non-Hermitian system. In each encircling loop, the diabolic point will be crossed once. In this way, an input mode can be converted into any desired mode by choosing a proper winding number (1 or 2 in this work) and a winding direction (i.e., clockwise or counter-clockwise).

OUR REPLY

This is an excellent summary of our results. We thank the Referee for carefully reading our manuscript.

REFEREE'S COMMENT 2

I will first talk about why in my opinion this work deserves publication although it is a theoretical work without experimental verification. The pioneering work on the dynamical encirclement of an exceptional point was published in [Nature 537, 76 (2016)], where the authors studied a two-state non-Hermitian system and a chiral mode switching phenomenon was found due to the non-adiabatic transitions between eigenmodes in non-Hermitian systems. Such breakdown of adiabaticity cannot be avoided in the dynamical process so that even multi-state systems have been studied in various works, we still cannot realize mode switching on demand as that in the stroboscopic case. After studying this work, I realize that it may be because that previously we focus on an encircling process in which two system parameters are changing and therefore we have an unchanged energy surface. In this work, the authors introduced a third varying parameter (i.e., the term “g”) so that the energy surface is also changing when the state evolves on it. In this way, by introducing a diabolic point which can connect two “sub-systems”, the varying energy surface allows a desired mode switching functionality. I believe that such mode switching behavior cannot be realized using previous non-Hermitian design principles. I think this work adds something very important and interesting to this research field and the design idea (i.e., a varying energy surface) will inspire researchers to find new non-Hermitian physics and applications. Therefore, I would like to support the publication of this work and I give some comments below for the authors' consideration.

OUR REPLY

We are very pleased by the Referee's appreciation of the article, and we thank again the Referee for the very detailed reading of our manuscript.

REFEREE'S COMMENT 3

The system proposed in this work is quite complex compared to previous systems with an unchanging energy surface. Therefore, it would be better to add some discussions on the non-adiabatic transitions. I assume there are at most two non-adiabatic transitions, e.g., in a process where both the initial and final mode are ψ_1 . Since different switching processes can have different number of non-adiabatic transitions, such information may help the readers better understand the underlying physics.

OUR REPLY

We thank the Referee for that suggestion. To elaborate on the effects of non-adiabatic transitions (NATs) and crossing the DC, in the revised version of the manuscript, we modified Fig. 4 and incorporated a new Fig. 5. The modified Fig. 4 now illustrates the fidelity between the time-evolving state $\psi(t)$ and the NHH eigenstates ψ_k at the the same moment of time. This can help to shed light on the system dynamics and various NATs which occur during the state evolution. The new Fig. 5 is the representation of panel (a) in Fig. 4, but in the energy space. We now also discuss and comment in detail on NATs in the main text of the revised manuscript (subsection "Winding an exceptional curve by crossing diabolic points").

REFEREE'S COMMENT 4

The system is a PT-symmetric system with gain and loss. What about the phenomenon in a passive system with loss only, as such passive system may be realized in potential experiments. As there may be multiple non-adiabatic transitions in the proposed system, i.e., the state has more chances to evolve on the higher-loss energy sheet, the transmission may be very low for potential applications.

OUR REPLY

Our results indicate that the considered mode switching mechanism can still be observed even for loss-only systems without gain. To elaborate more on this, in the Supplementary Note

1, we now discuss the mode swapping for the case of the two neutral and two lossy cavities. Our calculations imply that with decreasing gain, the system becomes more susceptible to the system parameters. That is, more constraints are imposed on the winding time and mode coupling. Moreover, the transmission coefficient also decreases with time. However, injecting gain, small enough to keep the system linear, substantially improves the mode switching functioning in terms of its resilience to perturbations and signal transmission.

REFEREE'S COMMENT 5

What symmetry governs that the system always has two energy levels with the same lowest loss (or highest gain)?

OUR REPLY

The perturbed system is additionally governed by the chiral symmetry which is responsible for the given energy distribution. We have added a new paragraph below Eq. (7) to explain that fact:

The energy spectrum of $H(\delta)$ consists of two pairs of Riemann sheets. For real-valued energies, these pairs may or may not intersect, depending on the system parameters, as shown in Figs. 2(a) and 3. For imaginary-valued energies, on the other hand, these pairs always coincide, as follows from Eq. (4) [see also Fig. 2(b)]. Though the chosen perturbation lifts the \mathcal{PT} -symmetry of the NHH in Eq. (3), the NHH $\hat{H}(\delta)$ still possesses the chiral symmetry $\mathcal{C}\hat{H}(\delta)\mathcal{C}^{-1} = -\hat{H}(\delta)$, where \mathcal{C} is the Hermitian operator satisfying $\mathcal{C}^2 = 1$, expressed via the antidiagonal matrix $\mathcal{C} = \text{antidiag}[1, -1, -1, 1]$. For this chiral symmetry one always has: $E_k = -E_{5-k}$, with $k = 1, \dots, 4$ (see Fig. 2).

REFEREE'S COMMENT 6

In a conventional system where “g” is a constant, the proposed programmable switching functionality cannot be realized, since given an initial state and winding direction, one winding loop and two winding loops will result in the same final mode. The authors may add some discussions or a table on this point, which can emphasize the role of the diabolic point in realizing the phenomenon and also help readers intuitively understand the breakthrough of this work.

OUR REPLY

We thank the Referee for this suggestion. We have added two new equations [namely Eqs. (9) and (10), in the revised manuscript] to highlight the fundamental role of the diabolic point in the system dynamics. Table 1 combined together with Eqs. (9) and (10) reveal the possibility of realizing a programmable symmetric-asymmetric mode switching in the \mathcal{PT} -symmetric four-mode bosonic system, in which either by crossing or not crossing the DC, one can control the chiral mode behavior in the system. We comment on this in more detail in the main text of the revised manuscript.

REFEREE'S COMMENT 7

In the current system, the term “ g ” is required to be zero at $\omega t = \pi$, in order to reach the diabolic point. What happened if “ g ” is very small but not zero at that point? Can we still have the same mode switching phenomenon?

OUR REPLY

Our analysis shows that the mode switching considered is preserved even when introducing infinitesimal perturbations to the coupling g at times when the state is supposed to traverse the DC. This can be understood as the diabatic evolution of the state in the vicinity of the DC, which enables the state to ‘jump’ to another energy surface even though the state does not cross the DC. We have also added the corresponding paragraph in the Discussion section to explain that.

Our analysis shows that the mode switching presented is resilient to various forms of perturbations. For instance, as Fig. 4 indicates, changing the starting point of winding does not affect the results. The same applies when perturbing gain, loss, or mode coupling. Moreover, the system is robust to small perturbations in the mode coupling $g \rightarrow g + \epsilon$, for $\epsilon \ll g$, at times $t = T/2, T$. The latter fact can be understood as the diabatic evolution (on the scale of ϵ) of the state in the vicinity of the DC, which enables the state to transfer to another energy surface even though the state does not cross the DC.

***** REPORT OF THE REFEREE 2 *******REFEREE'S COMMENT 1**

The manuscript presents theoretical studies of a programmable multimode switch based on dynamically encircling an exceptional curve while crossing diabolic points in non-Hermitian systems. A distinguishing aspect here is the proposed-a new programmable mode switch at a four-states PT-symmetric system with a diabolically degenerate exceptional point by breaking adiabaticity. The topic is interesting to broad communities of theoretical and applied physics and photonics researchers dealing with PT-symmetric systems.

OUR REPLY

We thank the Referee for the summary and the overall positive comment of our work. We appreciate the time that the Referee spent on carefully reading our manuscript.

REFEREE'S COMMENT 2

There are a number of questions and comments

OUR REPLY

Below, we address the Referees questions and comments step-by-step.

REFEREE'S COMMENT 3

Given that the fabrication imperfections are omnipresent in the experiment, how is such a system robust for coupling, gain, and loss perturbation? Such information definitely will elevate the work.

OUR REPLY

Our calculations show that the system remains robust against various forms of perturbations occurring either in the gain, loss, or mode coupling. For that, we have added a new paragraph

in the Discussion section:

Our analysis shows that the mode switching presented is resilient to various forms of perturbations. For instance, as Fig. 4 indicates, changing the starting point of winding does not affect the results. The same applies when perturbing gain, loss, or mode coupling. Moreover, the system is robust to small perturbations in the mode coupling $g \rightarrow g + \epsilon$, for $\epsilon \ll g$, at times $t = T/2, T$. The latter fact can be understood as the diabatic evolution (on the scale of ϵ) of the state in the vicinity of the DC, which enables the state to transfer to another energy surface even though the state does not cross the DC.

Moreover, our analysis also shows that the mode switching mechanism studied here can be observed even in passive \mathcal{PT} -symmetric systems, where the gain is much lower than the loss or even absent in the system. We present our results on the latter in the Supplementary Note 1.

REFEREE'S COMMENT 4

The authors claim that the results for the four-states system can be extended to any even-multimode \mathcal{PT} system with a set of m second-order ECs accompanied by the corresponding pairs of DCs. In this respect, and given that the work is purely theoretical, it would be beneficial to present an example of the higher-states system in supplementary materials.

OUR REPLY

We followed the Referee's suggestion and we extended our results to an eight-mode system, which we now present in the Supplementary Note 2.

REFEREE'S COMMENT 5

The photonics platform seems to be the perfect candidate to realize such a switch. Can authors propose one of the specific photonic systems and physical parameter ranges for such realization?

OUR REPLY

For the experimental realization of the multimode switch, one can use coupled toroidal microcavities, or, for instance, a set of coupled waveguides. The system of four coupled

whispering-gallery mode microcavities with gain and loss can be fabricated in the manner presented in Ref. [9] in the revised manuscript, whereas the photonic setup based on coupled waveguides can be implemented based on the scheme shown in Ref. [54]. We have added such an explanation in the ‘Discussion’ section.

***** REPORT OF THE REFEREE 3 *******REFEREE'S COMMENT I**

In this work, the authors describe a tuneable switch between 4 modes with gain and loss, which relies on the existence of certain features (diaboloic points and exceptional curves (EC)) in the complex spectrum of those modes. In particular, it is shown that under certain conditions, each of the initial eigenmodes can be mapped onto any other mode by encircling the EC once or twice and clockwise or counter-clockwise.

OUR REPLY

This is a good summary of our results. We thank the Referee for carefully reading the manuscript.

REFEREE'S COMMENT II

While this analysis extends previous works on encircling exceptional points (EP) and will be of interest for the targeted audience, I have several reservations about a publication of this manuscript in a broad and high-impact journal such as Nature Communications:

OUR REPLY

In the reply letter and revised version of the manuscript, we address all the comments and critique raised by the Referee. We believe, that in its current revised form, the manuscript is now suitable for publication in Nature Communications.

REFEREE'S COMMENT 1

Over the past 5 years there has been a considerable interest in the dynamical encircling of EP and in the resulting asymmetric mode switching arising from a break-down of adiabaticity in non-Hermitian systems. These effects are somewhat counterintuitive when viewed from the perspective of Hermitian quantum mechanics, but can be easily understood in terms of coupled modes with varying gain and loss coefficients. Also, in contrast to what is often claimed in the literature, the existence of exceptional points or other 'topological' features in the eigenvalues spectrum is not essential to observe these effects. While encircling an

EP leads to the correct alternation between loss, gain and coherent interactions, the same evolution can be obtained by simply switching on and off these contributions in the correct order.

In the current example, the existence of a diabolic point (which one could also simply call a level crossing) switches off the coupling between some modes, which would otherwise be mixed and lead to a more complicated evolution. This makes it convenient to achieve a desired coherent mixing or non-mixing of modes, but it doesn't introduce a new fundamental principle.

OUR REPLY

We would like to address this critique.

First of all, even when considering a mode switching mechanism occurring in systems with no EPs, as the Referee pointed out, it is questionable whether it is possible at all to control various symmetric and asymmetric mode switching combinations when there are many modes involved.

Second, according to Refs. [61,62] in the revised manuscript, in systems with EPs, but when winding trajectories do not encompass EPs, a chiral (asymmetric) mode conversion can still be realized. However, such results have been shown only for two-mode systems and they are valid only for winding contours in the very close vicinity of EPs (see Ref. [61] and its Erratum). This also imposes additional constraints on versatility and scalability of such a method, when extending it to multimode systems.

Whereas in our work we propose a fully controlled and programmable *symmetric* and *asymmetric* mode switching mechanism in genuine *multimode* systems. Furthermore, in order to show that our results can be straightforwardly extended to any multimode systems, in the Supplementary Note 2, we additionally implement a proposed symmetric-asymmetric mode switching for an eight-mode system.

We also note that a correct alternation between loss, gain and coherent interactions for a four-mode system with fixed interconnected low-order EPs, as was studied, e.g., in Ref. [52], does not allow to achieve the same (a)symmetric mode-transfer universality as demonstrated in our work. For instance, in our work, one can asymmetrically switch to any desired mode irrespective of the initial state. On the contrary, in Ref. [52], to ensure that the system will always end up in a given state requires the knowledge of the initial state because the mode coupling alternations are state-dependent. Moreover, our proposed scheme also enables to restore a system's state-swap symmetry in the $n \geq 2$ cycles for a given winding direction, which is difficult to achieve in Ref. [52]. This symmetric-asymmetric mode switching universality in multimode systems is what fundamentally distinguish our results from previous works.

Our findings, thus, shed new light on unique properties of multimode systems possessing both types of degeneracies and can be utilized in novel light manipulation protocols. To stress that, we have also modified the title and abstract, and substantially rewritten the main text in the revised version of the manuscript.

REFEREE'S COMMENT 2

Identifying a way to achieve a tuneable four-mode switch through the encircling of EC is an interesting academic exercise, but doesn't seem to have any practical value. There are much simpler ways to realize the same capabilities, for example, by turning on and off coherent couplings. The current dissipative implementation has the additional drawback, that it cannot be used for quantum information processing or low-power classical applications.

OUR REPLY

Again, as we have already stressed in the previous reply, even though there may be a simpler way to induce mode switching (e.g., by time-dependent coupling), in this case one would not achieve such a (a)symmetric universality. On the contrary, no matter the state we begin with, after our proposed protocol (encircling the DDEP) we *always end up in a given desired state*.

In order to show that the system does not have drawback when used in low-power classical applications, we additionally present numerical simulations for a passive \mathcal{PT} -symmetric system with or without low gain, in the Supplementary Note 1. Our calculations indicate that the mode switching mechanism is preserved for such systems, which can be exploited in low-power classical applications as well as for quantum protocols based on postselection and/or dilation procedures. We added the corresponding text in section 'Discussion' in the revised manuscript.

REFEREE'S COMMENT 3

The discussion around Fig. 3 is extremely difficult to follow and it is very hard to get any intuition out of it. It might be a lot easier to understand the evolution by plotting, for example, the real parts of the eigenvalues along the parameter trajectory, $Re[E_k(t)]$, and indicating the instantaneous loss/gain rates of each eigenstate by its color or similar means. In any case one should think about how to make this discussion more clear.

OUR REPLY

We thank the Referee for this suggestion. To elaborate on the system dynamics in more detail, in the revised manuscript, we have modified Fig. 4 and added a new Fig. 5. The Fig. 4 now illustrates the fidelity between the time-evolving state $\psi(t)$ and the Hamiltonian eigenstates ψ_k at time t . This provides information on the system non-adiabatic transitions and the state's trajectories over time. The new Fig. 5 represents the state's evolution in real energy space, which corresponds to panel (a) in Fig. 4. To explain in more detail the system

dynamics, presented in Figs. 4 and 5, we have also added new text in the revised manuscript (see new subsection “Winding an exceptional curve by crossing diabolic points”).

REFEREE’S COMMENT 4

There is no discussion about the imaginary part of the eigenvalues, which is responsible for the non-adiabatic transitions between the eigenvalues sheets. In the encircling of a single EP, this is the most essential part in the dynamics, since the system will always end up in the mode with maximal gain (or minimal loss). What is the role of non-adiabatic transitions in the current setting or how can it be that these non-adiabatic effects don’t appear?

OUR REPLY

Following our previous reply, in order to reveal the role of imaginary parts of the eigenvalues on the non-adiabatic transitions (NATs), we have modified Fig. 4 and added a new Fig. 5. We now also discuss the nature of NATs in the subsection “Winding an exceptional curve by crossing diabolic points”. There, we reveal that imaginary parts of the eigenvalues play the same role on the occurrence of the NATs as in the case of independent two two-mode systems, which has been already elaborated in detail in e.g., in Ref. [54].

REFEREE’S COMMENT 5

Related to 4), in the plots in Fig. 4, all the overlaps between the time-evolved wavefunction and the initial eigenmodes are of order 1. However, during the evolution, the different modes experience gain and loss and one expects that the amplitudes are amplified or damped by many orders of magnitudes. Why does this not show up in these plots?. How sensitive is the switching behavior with respect to variations in ϕ_0 and T ?

OUR REPLY

For better readability, in Fig. 4 (in the previous and modified version of the manuscript) we plotted the fidelity for normalized states, though the evolving states indeed become unnormalized in the broken \mathcal{PT} -phase. Since we are interested in the final states in the exact \mathcal{PT} -symmetric phase, the information about unnormalized behaviour would only complicate the explanation of the system dynamics, as presented in Fig. 4. To stress this point, we have also added an explanatory sentence in the caption of Fig. 4.

LIST OF CHANGES MADE

- The title has been modified.
- The abstract has been substantially modified.
- The last paragraph of the ‘Introduction’ section has been changed.
- The main text in the ‘Results’ and ‘Discussion’ sections has been completely rewritten.
- Figs. 3 and 4 have been modified.
- A new Fig. 5 has been added.
- The list of references has been reduced.
- New Supplementary information has been provided.

Since the main text in the revised manuscript has been substantially modified in all sections, except the Introductory part, we do not mark or highlight the changed text there.

REVIEWERS' COMMENTS

Reviewer #2 (Remarks to the Author):

The authors have addressed all my concerns in a very satisfactory way. I am very happy now to support its publication.

Reviewer #3 (Remarks to the Author):

I am satisfied with the response.

In conclusion, I support the manuscript for publication in Nature Communications.

Reviewer #4 (Remarks to the Author):

In my previous report I questioned the overall impact / generality of this work and also pointed out several important aspects that were unclear or lacked important clarifications.

In their reply, the authors addressed all my points of critics in detail and made substantial revisions to the manuscript. In particular the newly added Figures and the revised explanations make it much easier to understand the evolution of the system, including the non-adiabatic transitions. I appreciate the effort made by the authors, which considerably improved the readability of the paper and removed all my "technical" concerns. While still question the broader impact of this technique, the paper is certainly of interest for the field of non-Hermitian physics as also confirmed by the other referees. Therefore, I recommend a publication of the revised version of this manuscript in Nature Communications.